# Influence of Hydrothermal Treatment of Brewer’s Spent Grain on the Concentration and Molecular Weight Distribution of 1,3-1,4-β-D-Glucan and Arabinoxylan

**DOI:** 10.3390/foods12203778

**Published:** 2023-10-14

**Authors:** Julia Steiner, Michael Kupetz, Thomas Becker

**Affiliations:** Research Group Beverage and Cereal Biotechnology, Institute of Brewing and Beverage Technology, Technical University of Munich, 85354 Freising, Germany

**Keywords:** brewer’s spent grain, BSG, 1,3-1,4-β-D-glucan, β-glucan, arabinoxylan, AX, hydrothermal treatment, molecular weight, molecular weight distribution, thermal degradation, HMF, furfural

## Abstract

Brewer’s spent grain (BSG) is the most abundant residual in the brewing process. Non-starch polysaccharides such as 1,3-1,4-β-D-glucan (β-glucan) and arabinoxylan (AX) with proven beneficial effects on human health remain in this by-product in high amounts. Incorporating the valuable dietary fiber into the food industry could contribute to a healthy diet. However, a major challenge is extracting these dietary fibers (i.e., β-glucan and AX) from the solid residue. In this study, hydrothermal treatment (HT) was applied to dissolve the remaining water-insoluble carbohydrates from BSG with the aim to extract high amounts of β-glucan and AX. Particular focus was placed on the molecular weight (MW) range above 50 kDa and 20 kDa, respectively, as these are considered to have health-promoting effects. Different treatment temperatures, reaction times, and internal reactor pressures were tested to determine the best process settings to achieve high yields of β-glucan and AX and to examine the influence on their molecular weight distribution (MWD). Overall, 85.1% β-glucan and 77.3% AX were extracted corresponding to 6.3 g per kg BSG at 160 °C and 178.3 g kg^−1^ at 170 °C, respectively. However, less than 20% of both fiber substances were in the desirable MW range above 50 kDa and 20 kDa, respectively. When lower temperatures of 140 and 150 °C were applied, yields of only 3.0 g kg^−1^ β-glucan and 128.8 g kg^−1^ AX were obtained, whereby the proportion of desirable fiber fractions increased up to 45%. Further investigations focused on the heat-induced degradation of monosaccharides and the formation of undesirable by-products (i.e., HMF and furfural) that might pose a health risk.

## 1. Introduction

Brewer’s spent grain (BSG) is the major by-product during beer production. Although it is mainly used for animal feeding, sustainability concepts and growing health awareness have made BSG an attractive raw material for high value-added applications in the food industry [1,2]. BSG contains an adequate amount of non-starch polysaccharides, such as 1,3-1,4-β-D-glucan (β-glucan) and arabinoxylan (AX), which provide extra nutritional value [2]. However, a heterogeneous structure and previous steps during the brewing process make it difficult to solubilize the fiber components [3]. In cereals, β-glucan is a linear and unbranched polysaccharide composed of β-D-glucopyranosyl residues. The rigid (1–4)-sequences are irregularly broken by (1–3)-linkages [4,5] that affect solubility behavior of β-glucan in water [6,7]. AX consists of (1–4)-linked β-D-xylopyranosyl residues forming the backbone. Selected xylose residues can be substituted with varying amounts of arabinose, which are potentially substituted with ferulic acids, which are susceptible to oxidative cross-linking [1,8,9,10]. The ability to form gel networks and thus increase viscosity is an important physico-chemical property with beneficial effects on human digestion that supports potential physiological effects [11,12]. Especially, barley β-glucan and wheat AX are of interest, since in 2012, both were registered on the positive list of the European Food Safety Authority (EFSA), as ingredients that have beneficial effects on human health under certain conditions [13]. Although these are indigestible by human digestive enzymes [14,15], both non-starch polysaccharides can lower the postprandial glycemic response and insulin levels in blood. These responses are important benefits with regard to diabetes [16,17]. The literature suggests that the molecular weight (MW) of the fiber must exceed a certain value to increase the viscosity of the digestive effluents. Several authors even recommend that the MW should be above 50 kDa for β-glucan and 20 kDa for AX to affect postprandial glycemic and insulinemic responses significantly [18,19,20]. Moreover, β-glucan can reduce elevated blood cholesterol levels, one of the main risk factors for coronary heart diseases [21]. It was also shown that β-glucan always had the same effect on plasma cholesterol levels in mice despite different MW ranges [22]. By contrast, latest studies about in vitro fermentation of β-glucan using human fecal microbiota have shown that an average MW of 28 kDa appears most promising for the development of a potential new prebiotic [23]. Similarly, AX has gained interest as a new class of prebiotics, leading to enhanced immune function in the host [24,25]. However, the rate of fermentation and the generated metabolites heavily depend on the MW and the degree of polymerization [19]. Recent findings from Neyrinck et al. [26] suggest that AX of relatively low MW may serve as a promising prebiotic nutrient as it improved obesity-derived metabolic disorders such as type 2 diabetes, hypercholesterolemia, and fatty liver disease. Based on the mentioned specifications of the EFSA and various scientific studies, the respective health claims refer to a defined MW range that is recommended to achieve significant results. Therefore, it is not primarily the total yield of β-glucan and AX, but the percentage in the individual MW ranges that is decisive.

To release valuable fiber from BSG without major degradation, auto-hydrolysis under mild conditions should be applied. Hydrothermal processing based on liquid hot water (LHW) has distinctly different solvent properties compared to water under ambient conditions [27]. LHW has already been used as a pretreatment process for saccharification and ethanol production of cellulose and lignocellulosic material [28] as well as the extraction and fractionation of high MW β-glucan from different cereals (e.g., barley or wheat) [29]. Benito-Román et al. [29] and Carvalheiro et al. [30] demonstrated that easily extractable cereal compounds, such as glucose, dextrin, and starch, were dissolved in water at temperatures from 110 to 160 °C. Non-starch polysaccharides such as β-glucan showed the highest extraction yield at temperatures between 150 and 160 °C (pH 5.7). When exceeding 160 °C, the β-glucan content declined, while glucose and pentose sugar concentrations noticeably increased. A further rise in temperature caused advanced decomposition of carbohydrates (pH 4.2), leading to the formation of corresponding heat-induced sugar degradation products such as HMF and furfural. Similar results were observed during hydrothermal treatment (HT) of pure β-glucan and AX model solutions, showing thermo-labile behavior when temperatures rise beyond 160 °C [31]. According to the literature, treatments above 140 °C initially affect the tertiary structure by causing disordered chain conformation that is irreversible when reaching 155 °C [32]. A rise in temperature up to 180 and 200 °C showed a significant decline in MW and dispersity. Temperatures above 200 °C additionally caused advanced saccharification and an intensified formation of 5-hydroxymethylfurfural (HMF) and furfural [31].

Several investigations have shown that hydrothermal processing could be a versatile, innovative, and environmentally friendly technology for transferring insoluble carbohydrates from fiber-rich raw or residual material into the aqueous phase [27,28,29,30]. Currently, there are several studies on the solubilization of polysaccharides, the production of oligosaccharides, as well as the saccharification and further processing by means of HT (i.e., auto-hydrolysis). However, none of the research has focused or referred to the extraction of specific MW ranges of dietary fibers from BSG, which might be associated with possible beneficial effects for human health. The combined influence of treatment temperature and reaction time on both the extraction yield and structural properties of β-glucan and AX are highly auspicious for targeted separation of individual MW ranges.

The purpose of the present study was to apply HT to BSG and determine the impact of three variable parameters (i.e., internal reactor pressure, treatment temperature, and reaction time) on the yield and molecular weight distribution (MWD) of β-glucan and AX. In addition, the hydrolysis conditions required to produce the maximum amount of solubilized fiber in a specific MW range (>50 kDa for β glucan and >20 kDa for AX) while simultaneously minimizing the fragmentation, saccharification, and formation of further heat-induced degradation products (i.e., HMF and furfural) were determined.

## 2. Materials and Methods

### 2.1. Brewer’s Spent Grain and Chemicals

Barley BSG was kindly provided by a German brewery (β-glucan: ~1% (dry weight) and AX: ~23% (dry weight)). The fresh BSG was immediately dried to minimize the risk of microbial spoilage. A gentle temperature of 70 °C was chosen for circulating air-drying; high enough to reach a moisture content of 3% within four hours, but low enough to prevent the formation of thermally generated compounds. After drying, BSG was vacuum packed and stored in a dry and dark location until further analysis. Before hydrolysis, BSG was ground to a fine powder (0.5 mm) using a hammer mill (Retsch GmbH, Haan, Germany).

Barley β-glucan (1,3-1,4-β-D-glucan, medium viscosity, *M_w_*: 251 kDa, *M_w_*/*M_n_*: 1.96, purity: ~95% (dry weight basis)) and wheat AX (P-WAXYM, medium viscosity, *M_w_*: 323 kDa, purity: ~95% (dry weight basis), arabinose: 38%, xylose: 62%) were purchased from Megazyme (Bray, Ireland). D-(+)-xylose (X1500, purity: ≥99% (dry weight basis)) was purchased from Sigma-Aldrich (Darmstadt, Germany).

### 2.2. Hydrothermal Treatment

#### 2.2.1. Equipment for HT

A pilot-scale high-pressure plant (p_max_: 345 bar, T_max_: 350 °C, V_max_: 1200 mL) from Parr Instrument Company (Moline, IL, USA) was used. The vessel was equipped with a stirrer (speed: 600 min^−1^) to ensure homogeneous mixing of the suspension and to minimize internal temperature gradients. For controlling the pressure and increasing it if necessary, an external nitrogen reservoir was used, which was connected to the vessel interior via an interface in the lid. An electric heating jacket was used to achieve and maintain the target temperature in the vessel. Heating power was controlled manually depending on the latest temperature in the vessel. Internal reactor pressure and temperature were continuously measured and recorded. Sampling was performed in the pressurized condition via a sampling valve connected to a special sample container for pressure relief.

#### 2.2.2. HT Parameters

The experimental design investigated the effect of internal reactor pressure, treatment temperature, and reaction time on the yield and molecular weight distribution of β-glucan and AX recovered from BSG. To cover a broad range of HT variation, three different pressure levels (50, 100, and 150 bar) were manually set and tested. However, for most treatments, the pressure adjusted autonomously depending on the temperature in the closed system. The treatment temperature was tested at seven different levels, ranging from 130 to 200 °C with 10 °C increments. By collecting the samples after 2, 5, 10, 15, 20, and 30 min, it was possible to evaluate the interaction of temperature and time on β-glucan and AX yields.

#### 2.2.3. BSG Sample Preparation and Application of HT

To prepare the suspension, 30 g of ground BSG were manually mixed with 1000 mL of tap water. The solid concentration (i.e., 30 g) was selected from preliminary experiments; this is the maximum concentration that does not block the exhaust valves of the high-pressure equipment. After filling the vessel, it was sealed and heated to the target temperature and kept constant for the duration of the treatment. After the selected reaction end time, 150 mL of the produced hydrolysate were collected and divided into three 50 mL tubes for faster cooling to room temperature. Immediately afterwards, the BSG hydrolysates were frozen and finally freeze-dried to reduce the moisture content (<1%). Samples were stored at −20 °C until further analysis.

### 2.3. BSG Hydrolysate Characterization

To characterize the produced BSG hydrolysates, the freeze-dried samples were resuspended in double distilled water. The suspensions containing 3% (*w*/*w*) solids were first heated for 20 min in a water bath not exceeding 80 °C and then homogenized for 1 min to ensure thorough mixing. After cooling to room temperature, samples were gently centrifuged to achieve a faster sedimentation of the particles (2000 rpm, 5 min). As indicated for each method of analysis, the suspensions (3% *w*/*w*) were either concentrated by gentle heating or diluted with double-distilled water.

### 2.4. Quantification of 1,3-1,4-β-D-Glucan Using an Enzymatic Assay

Concentration of β-glucan was determined following Analytica EBC method 8.13.1, consisting of an enzymic procedure for the quantitative measurement of 1,3-1,4-β-D-glucan according to McCleary and Codd [33]. This method is based on enzymatic reactions caused by highly purified lichenase (endo-1,3:1,4-β-D-glucanase) from *Bacillus subtilis* (activity: ~250 U/mg (40 °C, pH 6.5 on barley β-glucan); concentration: ~1000 U/mL) and β-glucosidase from *Aspergillus niger* (activity: ~90 U/mg (40 °C, pH 4.0 on p-nitrophenyl β-glucoside); concentration: ~40 U/mL). All samples were suspended and hydrated in a sodium phosphate-buffered solution (pH 6.5) and then incubated with lichenase. After filtration, an aliquot was hydrolyzed to completion with purified β-glucosidase and the glucose produced was determined colorimetrically. All enzymes and chemicals were purchased from Megazyme (Bray, Ireland). The absorbance was read at 510 nm using a spectrophotometer (DR5000; Hach Lange GmbH, Düsseldorf, Germany). Measurements were performed in triplicate (*n* = 3).

### 2.5. Quantification of Water-Extractable AX Using a Colorimetric Assay

AX concentration was determined using a colorimetric assay for quantification of water-extractable AX according to Steiner et al. [34]. This method is based on the interaction between degradation products of pentose sugars obtained by acid hydrolysis and the dye phloroglucinol [35]. For the calibration curve, a dilution series of a xylose stock solution was used (0, 50, 100, 150, 200, 250, 300 mg L^−1^). Additionally, two AX control standards (250 and 500 mg L^−1^) were included in each measurement series as a control to determine the recovery rate. Sample and standard aliquots as well as the freshly prepared reaction reagent [34] were added to brown glass tubes, which were heated in a boiling water bath for 25 min. All chemicals were obtained from Merck (Darmstadt, Germany). Subsequently, the absorbance was read at 550 and 505 nm using a multi-mode microplate reader (BioTek Synergy H4 with Hybrid Technology; BioTek Instruments, Inc., Winooski, VT, USA). Measurements were performed in triplicate (*n* = 3 × 4).

### 2.6. Determination of Relative MWD Using Gel Permeation Chromatography (GPC)

The MW analysis was conducted by means of gel permeation chromatography using the method described by Hartmann et al. [36]. The setup comprises an ÄKTAprime plus chromatography system (GE Healthcare, Solingen, Germany) equipped with a Superdex 200 pg prepacked column (HiLoad 26/600; GE Healthcare, Solingen, Germany). A phosphate buffer (pH 7.0) obtained from Merck (Darmstadt, Germany).served as eluent at a flow rate of 2.2 mL min^−1^. Calibration of the separation column regarding the molecular masses contained in the collected fractions was performed according to Kupetz and Gastl [37] using defined molecular size standards from Megazyme (Bray, Ireland). BSG hydrolysates were produced according to Section 2.2.3. Since the initial sample volume is highly diluted during chromatographic separation, the solids concentration of each sample subjected to MW analysis was adjusted individually. Based on the previously measured total β-glucan and AX content in the respective hydrolysate, the suspensions contained between 3 and 15% (*w/w*) solids. All samples were filtered (0.45 µm) and an initial volume of 3.0 mL was manually injected. The data were analyzed using PrimeView 5.0 software (GE Healthcare, Solingen, Germany). Measurements were performed in triplicate (*n* = 3).

### 2.7. Quantification of HMF and Furfural Using HPLC

High performance liquid chromatography (HPLC) was used for the quantitative determination of HMF and furfural according to the modified method described by Zappalà et al. [38]. The system consisted of an UltiMate 3000 Autosampler, an UltiMate 3000 pump module (Thermo Fisher Scientific, Inc., Waltham, MA, USA), an UltiMate 3000 separation module (Thermo Fischer Scientific, Dionex, Germering, Germany) coupled with an UltiMate 3000 Diode Array Detector (Thermo Fisher Scientific, Inc., Waltham, MA, USA). As the stationary phase, a Luna C18 column (4.6 mm × 250 mm, 5-µm particle size, 100 Å; Phenomenex, Aschaffenburg, Germany) was used. The mobile phase consisted of Solvent A (1% (*v*/*v*) formic acid in water) and Solvent B (100% acetonitrile). All chemicals were obtained from Merck (Darmstadt, Germany). Samples were filtered (0.45 µm) and 50 µL was automatically injected. The flow rate was set to 1.0 mL min^−1^ for 12 min. All samples were analyzed at a detection wavelength of 270 nm. An external seven-point calibration was used as the basis for a linear regression model to quantify HMF and furfural. The data were analyzed using ChemStation Rev.B.04.02 software (Agilent Technologies, Waldbronn, Germany). Measurements were performed in triplicate (*n* = 3).

### 2.8. Statistical Analysis

The data were statistically evaluated using OriginPro 2015G (OriginLab Cooperation, Northampton, MA, USA). A one-way ANOVA followed by Fisher’s least significant difference test with a significance level (α) of 0.05 was performed to identify the significance of the differences between the respective values.

## 3. Results and Discussion

### 3.1. Initial β-Glucan and AX Concentration in Untreated BSG Solid and Suspension

To evaluate the impact of internal reactor pressure, treatment temperature, and reaction time on the solubility and decomposition behavior of β-glucan and AX in BSG, which are inextricable in water under ambient conditions, the initial concentration was determined. First, the β-glucan and AX contents of the original solid BSG were measured to obtain reference values for the recovery rate. With amounts of 7.4 g kg^−1^ (β-glucan) and 230.6 g kg^−1^ (AX), our findings are consistent with those reported in the literature [1,39]. Barley BSG includes 20–25% (*w*/*w*) of AX, whereas the level of β-glucan is considerably lower, amounting to approximately 1% (*w*/*w*). Next, BSG was suspended in water and the β-glucan and AX concentrations were measured before the suspension was subjected to HT (i.e., the BSG control sample). Low concentrations of 0.2 g kg^−1^ (β-glucan) and 2.1 g kg^−1^ (AX) were measured. It is known that BSG does not contain any or a large amount of soluble substances, since these have been extracted during mashing in the brewing process. The mashing regime usually aims to guarantee the best enzymatic activity for starch and protein solubilization, therefore these substances are absent in the collected BSG. The small amount that was nevertheless measured might be traced back to the dissolution of possibly agglomerated fiber particles when samples were prepared for the chemical and structural analysis as described in Section 2.2.3.

### 3.2. Impact of Internal Reactor Pressure during HT of BSG

Initially, the influence of the internal reactor pressure on the release of β-glucan and AX from BSG was evaluated. For the treatment, temperatures of 160 and 180 °C were chosen, as hereby, the highest β-glucan and AX yields were achieved, and adequate amounts of HMF and furfural were present. However, the effect of pressure on the physico-chemical properties of hot water is very limited compared to temperature, so that pressure plays only a minor role in the extraction process of the said non-starch polysaccharides from cereal-based products [29]. If the pressure is sufficient to keep water in the liquid phase at an elevated temperature, the dielectric constant decreases. Under these conditions, water behaves like certain organic solvents, which can dissolve a wide range of low- and medium-polarity compounds [40]. For that reason, the pressure was first set to 150 bar. However, Benito-Román et al. [29] demonstrated that no significant effect on the extraction yield was observed as a function of pressure. This is consistent with our results, where reducing internal reactor pressure from 150 bar to 100 and 50 bar showed no significant difference (*p* > 0.05) in the total yield of β-glucan and AX (see Table 1). Even with only an equilibrium vapor pressure, almost no differences were observed compared to a treatment at 150 bar. On the contrary, especially at a treatment temperature of 160 °C where an internal reactor pressure of approximately 10 bar was reached, β-glucan and AX yields were sometimes even higher. These results agreed very well with the findings of Kronholm et al. [41], who determined that the pressure is not crucial for the extraction and thus it is not necessary to apply high pressures. Therefore, in further experiments, the internal reactor pressure was regulated autonomously, not exceeding 20 bar.

In addition, different internal reactor pressures were investigated with respect to the MWD of β-glucan and AX in the collected BSG hydrolysates, due to contradictory statements in the literature [29,40]. In Figure 1, the treatment temperature (160 °C) and reaction time (10 min) remained constant, while the pressure varied between 50, 100, 150 bar, and the equilibrium vapor pressure (<20 bar). Similar to the total β-glucan and AX yield, no significant differences were found in the MWD of AX. Although the internal reactor pressure was raised, the concentrations in the respective mass ranges stayed almost the same (±1.8%). When calculating the yield of medium and high molecular AX (>20 kDa), samples treated with vapor pressure showed a slightly higher level (38.6 g kg^−1^). However, this can be attributed to an already higher total AX yield (Table 1) compared to a treatment at 50 bar (34.6 g kg^−1^), 100 bar (33.4 g kg^−1^), and 150 bar (34.2 g kg^−1^). Moreover, similar results were observed with respect to β-glucan yields. These findings demonstrate that within the used temperature range, the internal reactor pressure had no influence on the solubility and decomposition behavior of β-glucan and AX from BSG as well as no effect on the MWD, which is consistent with those of Benito-Román et al. [29]. In addition, variation in the internal reactor pressure showed virtually no influence on the formation of heat-induced degradation products during HT. Therefore, for the subsequent treatments, internal reactor pressure was regulated autonomously (<20 bar) and only temperature and time were varied.

### 3.3. Influence of HT of BSG on the Total Concentration of β-Glucan and AX

#### 3.3.1. Effect of Treatment Temperature and Reaction Time on β-Glucan Concentration

Treatment temperatures of 130 and 140 °C resulted in β-glucan yields ranging from 2.8 to 3.8 g kg^−1^ without showing significant differences, especially when increasing the reaction time to 30 min at 140 °C (Table 2). A further temperature rise of 20 °C caused a significant increase in the β-glucan yield, whereby at 160 °C, the maximum, 6.3 g kg^−1^, was already reached after the 2 min holding period. A prolongation of the reaction time up to 20 min led to a clear reduction in the β-glucan yield to 4.2 g kg^−1^. It has to be considered that in some cases, heating the suspension to the target temperature already took up to 55 min, a comparatively long period with regard to the selected reaction times used for the treatment ranging from 2 to 40 min. Within the heating-up period, β-glucan was probably partially degraded before starting the defined reaction time [31]. At 150 °C, the β-glucan concentration initially showed a significant increase to 5.4 g kg^−1^ when extending the reaction time from 2 to 20 min. This amount even exceeded the yield that resulted from a 20 min treatment at 160 °C (4.2 g kg^−1^). A further prolongation of the treatment at 150 °C up to 40 min resulted in a significant decrease in the β-glucan concentration (3.1 g kg^−1^). These results are consistent with the findings of Xu et al. [32] who described the conformational transition of β-glucan between 140 and 160 °C. Heating β-glucan molecules at 155 °C induces a dissociation and bend of chains. Thus, the aggregated β-glucan, consisting of more rigid chains aligned to each other at lower temperatures, is partially destructed into flexible coils consisting of individual chains and aggregates [32]. Rising temperature beyond 155 °C resulted in a predominance of the flexible individual chains. At 170 °C, the aggregates were almost completely disassociated into single chains with a moderate thermal degradation [42].

In addition to the temperature, the duration of the treatment played an important role as efficiently demonstrated with the obtained results. A time extension to 20 min at 150 °C resulted in an increase in the β-glucan yield, however, a further prolongation led to a decrease. Thus, the reaction time also seems to have an impact on the ratio of aggregated to individual β-glucan chains and its degradation. Xu et al. [32] showed that the disordered structures occurred mainly after an extensive 30 min treatment at 155 °C. This observation provides strong evidence that the destruction of hydrogen bonds is also related to the treatment time being co-responsible for a transition into individual flexible chains.

When increasing the treatment temperature beyond 160 °C, a moderate decline in the β-glucan yield was observed, which quickly dropped to zero when reaching 200 °C, as the covalent bonds were already broken. Similar results were observed during HT of the pure β-glucan model solution, where temperatures above 200 °C promoted advanced saccharification as well as the accumulation of degradation products [31,32]. Most of the β-glucan molecules seem to have already been degraded and partially converted into undesirable by-products (e.g., HMF) that might pose a health risk. Concerning a possible shortening of the heating time, the results suggest that process temperatures of 160 and 170 °C combined with very short reaction times (e.g., 2 min) could lead to even higher β-glucan yields. According to the findings of Benito-Román et al. [29], increasing the reaction time at 170 and 180 °C beyond 2 min resulted in reduced β-glucan concentrations.

#### 3.3.2. Effect of Treatment Temperature and Reaction Time on AX Concentration

Compared to β-glucan, AX showed a slightly different solubility behavior in dependence of temperature and reaction time (Table 2). Similar to β-glucan, no significant difference was observed between the treatment temperatures 130 and 140 °C. At these temperatures, a prolongation of the treatment up to 15 min only led to a slight increase in the AX concentration. However, when treatment temperatures were elevated, an extension in the reaction time resulted in substantial changes in the AX yield from BSG. At 150 and 160 °C, a strong positive correlation (r = 0.99) was found over the entire treatment period of 30 and 20 min, respectively. A temperature rise up to 170 °C resulted in a strong increase in the AX yield, peaking at 178.3 g kg^−1^ (15 min), while further increases to 180 and 200 °C resulted in lower yields. In agreement with the findings of Coelho et al. [39] and Steiner et al. [31], the relative amount of arabinose increased significantly as the applied temperature was raised up to 180 °C. At 170 and 180 °C, the AX concentration as a function of the reaction time showed a polynomial regression, whereas at 200 °C, a high negative correlation (r = −0.99) was recorded. Enhancing treatment temperatures above 200 °C caused an increased degradation of AX fragments, which led to saccharification and further degradation of arabinose and xylose units to heat-induced by-products. The results agreed well with the findings of Coelho et al. [39] that an increase in temperature from 190 °C to 210 °C resulted in a decrease in the monosaccharide recovery of about 15%. At 210 °C, almost 30% of the total arabinose and xylose content was lost, which they attributed to thermal degradation.

The collected data indicate that the interaction of treatment temperature and reaction time have a significant impact on the resulting β-glucan and AX concentrations in the BSG hydrolysates. Treatment temperatures ranging from 150 to 170 °C appear to be very promising in terms of the maximum possible extraction yield. However, not only are the non-starch polysaccharide concentrations of the hydrolysates of interest, but also their compositions. The MWDs of the solubilized material serve as indicators, and in addition, previous studies have identified β-glucan and AX fractions of interest, which are potentially beneficial for human health.

### 3.4. Influence of HT of BSG on the Molecular Weight Distribution of β-Glucan and AX

#### 3.4.1. Effect of Treatment Temperature and Reaction Time on MWD of β-Glucan

The total β-glucan concentration in untreated BSG solid material was determined to be 7.4 g kg^−1^ from which a maximum of 85% could be recovered in the produced hydrolysates when treated at 160 °C for 2 min. However, 54.2% of the β-glucan molecules were smaller than 10 kDa (see Table 3). Especially with regard to possible human physiological effects based on the specifications of the EFSA and various scientific studies, it is not the total yield that is decisive, but the proportion in the individual MW ranges. A quantitative initial assessment on the MWD was done in separate studies using single substance model solutions by determining the fragmentation and decomposition patterns of the individual substances. Based on those findings, the MWD was specified by grouping into low (<10 kDa), medium (10–100 kDa), and high (>100 kDa) molecular β-glucan [43]. The medium MW range was additionally divided into two groups, 10–50 kDa and 50–100 kDa. The range from 10 to 50 kDa is of interest based on the previously reported possible prebiotic properties of β-glucan [23]. Moreover, molecular weights >50 kDa may be assigned to the range relevant to the claimed health effects declared by the EFSA Panel on Dietetic Products [44].

The BSG hydrolysates produced by HT at lower temperatures showed a much wider MWD of β-glucan (see Table 3) compared to treatments at 160 and 170 °C (10 min). Successive elevations in treatment temperature by 10 °C starting at 140 °C led to a significant increase in low MW β-glucan (<10 kDa), while the proportion in the upper medium MW range (50–100 kDa) suffered a sharp decline. When reaching 160 °C, the β-glucan percentage in the high MW range (>100 kDa) even dropped towards zero. At 170 °C, more than 80% of the β-glucan molecules were already smaller than 10 kDa. Although the concentration of β-glucan >50 kDa considerably declined resulting in a notably narrower MWD, the total β-glucan yield significantly increased at a treatment temperature of 160 °C (see Figure 2A). Whereas the proportion of β-glucan in the range 10–50 kDa showed only a slight increase, a noticeable proportion was shifted to the very low MW range (<10 kDa). Although the concentration was lowest at 140 °C (3.6 g kg^−1^) and highest at 160 °C (5.6 g kg^−1^), most β-glucan in the upper MW range (>50 kDa) was obtained at 140 °C. When the reaction time was extended to 15 min, the MW marginally shifted to the lower ranges and the distribution became slightly narrower (see Table 3). While the proportion of β-glucan molecules <10 kDa hardly increased when treating at 140 °C over a period of 15 min, a clear decrease was observed in the range >100 kDa. These results suggest that a treatment prolongation at 140 °C only leads to a restrained fragmentation, whereby the β-glucan from the high MW range was spread evenly among the lower fractions. Due to the corresponding total concentrations, the proportion of β-glucan molecules above 50 kDa remained almost unchanged despite an extension of the reaction time at 140 °C (see Figure 2B).

According to Xu et al. [32], the molecular size and weight of β-glucan sharply decreased in the temperature range of 140–160 °C. This could be ascribed to the irreversible transition from stiff chains to individual flexible chains as a result of the breaking of the intra- and intermolecular hydrogen bonding in β-glucan that start to break at 140 °C. It was shown that at elevated temperatures, the destruction resulted in the transition of aggregates into single β-glucan chains, whereas the breaking of intra-hydrogen bonding led to the bend of chains to form flexible coils responsible for the decrease in molecular size. Due to a reduced stability caused by a disordered chain conformation, the solubility and decomposition behavior changed. Sato et al. [45] observed that the stiff conformation of β-glucan aggregates remained intact at temperatures below 140 °C. However, at 160 °C, only single flexible coiled chains existed. This suggests that most of the hydrogen bonds, which sustained the chain stiffness and the aggregation, were disrupted leading to the rapid decrease in the hydrodynamic volume as a result of the bend of chains and dissociation of aggregates [32]. Thus, the destruction seems also to be directly related to the shift in MW towards lower ranges, partly accompanied by a thermal degradation consequently resulting in a change in the total concentration. The 10 min treatment at 140 °C led to a maximum β-glucan concentration of 1.4 g kg^−1^ in the >50 kDa range, which additionally was also the highest concentration in the upper MW range with regard to all applied process settings. Further extensions of the reaction time to 20 and 30 min, at 150 °C, resulted in a much narrower distribution with MWs below 100 kDa (see Table 3). The significant increase in the β-glucan concentration in the low MW range (<10 kDa) may be traced back to an advanced decomposition as described by Xu et al. [32]. Although the total yield at 150 °C (10 min) was much higher than at 130 and 140 °C, the β-glucan concentration in the upper MW range (>50 kDa) was just two-thirds of the amount at 140 °C (10 min), reaching only 0.9 g kg^−1^ (see Figure 2A). The maximum yield in the range 10–50 kDa was reached at 160 °C after 10 min (1.7 g kg^−1^). Treatments at temperatures above 160 °C led to excessive fragmentation into mainly medium and low molecular components. In agreement with the findings of Steiner et al. [31], further decomposition from the destruction of covalent bonds occurred at treatments above 200 °C. At a treatment temperature of 200 °C, nearly all β-glucan was found in the low MW range suggesting an almost uniform distribution.

#### 3.4.2. Effect of Treatment Temperature and Reaction Time on MWD of AX

The total AX concentration in untreated BSG solid material was determined to be 230.6 g kg^−1^ from which a maximum of 77% could be recovered in the produced hydrolysates (170 °C/15 min). Similar to β-glucan, the MWD of AX varied by the HT parameters used. For AX, the molecular weight of interest is above 20 kDa. AX was grouped into low (<20 kDa), medium (20–100 kDa), and high (>100 kDa) molecular AX [46]. Considering the hydrolysate with the maximum total yield (178.3 g kg^−1^) produced at 170 °C/15 min, 85.7% was detected in the low MW range (see Table 4). In spite of the considerably high total yield, only 25.5 g kg^−1^ of AX molecules larger than 20 kDa were obtained. The highest amount of AX with MWs above 20 kDa resulted from a 30 min treatment at 150 °C, accounting for 55.8 g kg^−1^, whereby the total yield was only 128.8 g kg^−1^. The same hydrolysate also yielded only 3.7 g kg^−1^ of β-glucan with more than 96% in the MW range below 50 kDa. At temperatures between 130 and 150 °C and reaction times up to 20 min, more than half of the AX molecules were detected in the MW range >20 kDa. According to Carvalheiro et al. [47], milder hydrolysis conditions led to greater proportions of high molecular AX. Similar to β-glucan, the AX molecules in the hydrolysates that were produced at lower temperatures showed a clearly wider-ranging MWD in comparison to treatments at 170 and 180 °C (see Figure 3). Interestingly, the rise in temperature from 130 to 140 °C resulted in a significant decrease in the low MW range (<20 kDa), whereas the medium range (20–100 kDa) showed a significant increase in the AX proportion. In the high MW range (>100 kDa), only small changes were recorded. The amount of low molecular AX was significantly smaller at 140 °C than at 130 °C. However, an elevation in the treatment temperature of 20 °C from 130 to 150 °C merely resulted in a marginal increase of about 3% in the low MW range (<20 kDa) and a clear drop in the high MW range (>100 kDa). This trend continued, as further successive temperature rises of 10 °C to 160 and 170 °C were performed. Within the low MW range (>20 kDa), a significant increase was observed reaching proportions of 48.5% and more, while the high MW range (>100 kDa) steadily decreased and dropped towards zero when reaching 170 °C (20 min). The medium range (20–100 kDa) showed only slight differences in the percentage, experiencing a significant decrease when a treatment temperature of 170 °C was reached (see Table 4). Regarding the AX concentration in the grouped MW ranges, the area >20 kDa showed a significant growth when elevating the treatment temperature successively by 10 °C up to 160 °C. Exceeding this temperature level caused a clear drop to 11.1 g kg^−1^.

Similar to β-glucan, when the reaction time was extended, a shift in MW was observed towards the lower ranges, implying a slightly narrower MWD. A clear reduction was observed in the upper range (>100 kDa), while the proportion of β-glucan <50 kDa and AX <20 kDa considerably increased. Regarding the AX range in between (20–100 kDa), the time dependent correlation changed in accordance with the treatment temperature. At 130 °C and 140 °C, clear positive relationships were recorded with r = 0.81 and 0.93, respectively, while at 170 °C, the correlation was strongly negative (r = −0.99). In contrast to all other treatment temperatures, at 140 °C, the MWD of AX showed a parabolic shape with the highest proportion in the medium range (20–100 kDa). In particular, the reaction time was a very sensible parameter for HT, as an extension of a few minutes may result in a significant change in the MWD.

#### 3.4.3. Importance of Temperature/Time Interaction for the MWD of β-Glucan and AX at Comparable Total Concentrations

The amount of β-glucan and AX in each grouped MW range must be considered separately and independent of the corresponding total concentration. The MWD of β-glucan and AX in two BSG hydrolysates produced at different process settings were either comparable or completely different, although the total β-glucan and/or AX concentration was almost the same. The comparison of two BSG hydrolysates produced at 150 °C/20 min and 160 °C/5 min amounting to total β-glucan concentrations of 5.4 and 5.7 g kg^−1^, respectively, showed similar levels of the β-glucan yield in each MW range; although, completely different temperature/time combinations were used (see Table 5). Regarding the MWD, a slight divergence was observed only in the medium range. Even though the process settings were evidently different, the intensity of the treatment seemed to be equal, consequently leading to almost identical yields in the grouped MW ranges. When the temperature was increased by 10 °C, the reaction time had to be reduced by 15 min to achieve almost identical results. A second pair of BSG hydrolysates produced at very similar process settings to the previous pair (150 °C/15 min and 160 °C/2 min)and varying only in the reaction time, which was reduced by 5 and 3 min, respectively, showed again an almost identical MWD. However, the total β-glucan yield of both hydrolysates differed significantly amounting to 4.6 and 6.3 g kg^−1^, respectively (see Table 1).

In contrast, two other hydrolysates (140 °C/10 min and 150 °C/30 min) that were prepared at 140 and 150 °C with a reaction time difference of 20 min showed almost identical total β-glucan yields (3.6 and 3.7 g kg^−1^) but two completely different MWDs (see Table 5). Only the proportions in the lower medium range (10–50 kDa) were on a similar level, which was also reflected in the β-glucan yield amounting to 1.1 and 1.0 g kg^−1^, respectively. Despite an almost identical total β-glucan concentration, the treatment conditions had varying effects on the MWD and therefore on the absolute amount in the ranges of interest for human physiological effects. The decrease in molecular size leading to a shift of β-glucan to the low MW range is probably attributed to the breaking of hydrogen bonds when applying temperatures above 150 °C [32]. A treatment at 140 °C/10 min seemed to be better, due to greater β-glucan yields in the medium and high MW range.

Comparing the total AX concentration of the first pair of BSG hydrolysates produced at 150 °C/20 min and 160 °C/5 min, almost identical values were measured. Regarding the MWD, merely a slight change was observed in the range above 20 kDa, which was noticeable by a minor shift of a maximum of 1.8% within the medium (20–100 kDa) and high (>100 kDa) MW ranges. In contrast, the proportion of molecules below 20 kDa remained constant (±0.2%). Consequently, the AX yield did not change significantly within the two grouped MW ranges (<20 kDa and >20 kDa). This is consistent with the findings of the β-glucan measurements (Section 3.4.1). Finally, it could be confirmed that completely different process settings were able to show equal treatment intensities, whereby nearly identical yields were achieved. However, another pair of BSG hydrolysates with nearly the same total AX concentration (±2.2%) produced at 160 °C/10 min and 170 °C/20 min showed completely different MWDs. The increase in treatment temperature of 10 °C and the doubling of the reaction time led to a shift of 26.1% to the low MW range (see Table 5). This was consistent with the findings of Carvalheiro et al. [47], who observed that the MW progressively declined with stronger hydrolysis conditions and low molecular AX accumulated. To achieve a larger amount of AX molecules above 20 kDa, which could potentially amount to support certain health-promoting properties in the human body [18], the milder treatment at 160 °C/10 min was more efficient. It resulted in a significantly higher AX yield in the medium and high MW ranges amounting to 38.6 g kg^−1^, whereas only about 10% of the AX molecules had MWs above 20 kDa when stronger hydrolysis conditions were applied (170 °C/20 min).

When the MWD of β-glucan and AX in the produced BSG hydrolysates are taken into account separately, a treatment temperature of 140 °C seems very promising in view of a maximum possible β-glucan yield in the MW range above 50 kDa. However, temperatures ranging from 150 (15–30 min) to 170 °C (up to 5 min) were more appropriate for producing BSG hydrolysates with an AX composition able to meet the requirements for possible health-promoting effects. The collected data on β-glucan and AX showed the importance of first assessing both parameters independently and then collectively. However, in addition to the characterization of the non-starch polysaccharides, other parameters should also be considered to determine the quality of the produced BSG hydrolysates.

### 3.5. Influence of HT of BSG on the Formation of Heat-Induced Degradation Products

To determine the most suitable HT parameters for BSG treatment, a number of indicators were considered. In addition to the total β-glucan and AX concentration as well as their MWD, the concentration of undesirable by-products should be monitored. A high content of the heat-induced degradation compounds, HMF and furfural, in the BSG hydrolysates might pose a health risk. In general, the consumption of foods containing HMF does not raise safety concerns. Due to the absence of data on reproductive and developmental toxicity, it is difficult to obtain a tolerable daily intake of HMF. According to Shapla et al. [48] a tolerable daily intake of 132 mg per day for HMF based on a 40-fold safety factor was determined. The average daily intake reported by the FDA changes from 4 to 30 mg HMF per person per day [49]. In contrast, for furfural, an acceptable daily intake value of 0.5 mg/kg body weight was established by the EFSA in 2004 [50]. The measured concentrations of HMF and furfural indicate that the variability in total β-glucan and AX yield is directly related to the formation of undesirable degradation products (see Table 6). The HMF concentration in the untreated BSG control sample was 0.2 mg L^−1^, while the furfural content was under a detectable level. Treatment at 130 °C led to relatively small amounts of HMF (0.6 mg L^−1^) and furfural (0.1 mg L^−1^), their concentrations showed no significant differences in BSG hydrolysates treated up to 150 °C/15 min. When the temperature was held constant at 150 °C but the reaction time was further extended up to 30 min, a substantial increase in concentration to 17.7 and 3.0 mg L^−1^ of HMF and furfural, respectively, was recorded. As expected, a further successive temperature rise of 10 °C resulted in significantly higher values with HMF peaking at 39.0 mg L^−1^ (180 °C/20 min). The maximum furfural concentration amounting to 11.7 mg L^−1^ was determined at a lower temperature, 170 °C/30 min. Consequently, the reaction time only had a significant influence by an extension of at least 10 min, whereas the increase in treatment temperature had a noticeably stronger impact. An elevation in temperature to 200 °C resulted in a proportionally higher increase in furfural concentrations on average by a factor of 2.5 compared to HMF, as the latter was subjected to secondary decomposition [28]. Different researchers observed that a further degradation of HMF and furfural led to an increase in acid concentration [51,52], which is consistent with our pH measurement. Starting at 160 °C, the stepwise increase of 10 °C already showed a significant effect on the amount of degradation products at a holding period of only 5 min.

Moreover, it was observed that some BSG hydrolysates had nearly the same HMF or furfural contents, although these were produced using different temperature/time combinations. However, most of these hydrolysates had completely different β-glucan and AX concentrations, except for two, produced at the following process settings: 150 °C/20 min and 160 °C/5 min. The measured furfural concentration in both samples was 1.3 mg L^−1^ and the HMF measured concentrations were 6.8 mg L^−1^ (150 °C/20 min) and 6.3 mg L^−1^ (160 °C/5 min). Consequently, the treatment intensities of 150 °C/20 min and 160 °C/5 min appeared to be identical, leaving room for choosing the possibly more energy-efficient option of lower treatment temperatures. Overall, all four substances, β-glucan, AX, HMF, and furfural, should be taken into account to choose the best treatment parameters. High yields of total β-glucan and AX are desired, but these should also be within the desired MW range and low in heat-induced degradation products.

### 3.6. HT Parameter Selection for High Quality BSG Hydrolysates

In order to produce hydrolysates suitable for human consumption, in particular, concentrations of 0.5 mg furfural per kg body weight must not be exceeded [50]. Considering several process settings of HT that provided the highest yield of β-glucan with the desired MW range (>50 kDa), no significant differences in HMF and furfural concentrations were observed despite varying treatment intensities. Very small amounts of HMF (1.0 mg L^−1^) and furfural (0.1 mg L^−1^) were produced at temperatures of 140 °C and 150 °C, which were considered the most effective for obtaining β-glucan above 50 kDa. An extension of the holding times up to 15 min showed only a slight increase to 2.6 and 0.7 mg L^−1^, respectively. However, process settings used to reach high yields of AX above 20 kDa delivered considerably larger quantities of undesired degradation products. Temperatures of 150 and 160 °C, which were highly efficient in obtaining AX above 20 kDa, yielded HMF levels of 1.9 (150 °C/2 min) and 6.1 mg L^−1^ (160 °C/2 min) as well as furfural levels of 0.3 (150 °C/2 min) and 1.2 mg L^−1^ (160 °C/2 min). Especially when the holding time was extended beyond 15 min, both products showed a significant increase up to 17.7 (HMF) and 3.0 mg L^−1^ (furfural). Even when treated at 170 °C, large quantities of AX above 20 kDa could still be obtained. However, the high amounts of HMF and furfural may lower the quality due to the associated health risks. During the first 10 min of the treatment, the undesired products showed only a slight increase from 8.4 to 10.7 mg L^−1^ (HMF) and from 1.6 to 2.3 mg L^−1^ (furfural), while a further prolongation to 30 min resulted in a significant rise up to 37.8 and 11.73 mg L^−1^, respectively.

## 4. Conclusions

The collected data show that hydrothermal treatment under controlled conditions is an effective tool to release inextricable non-starch polysaccharides from BSG. The extracted substances may have potential applications as a functional ingredient in the food industry. Based on the collected data, it was concluded that the interaction of the process settings contributes to the solubilization of high amounts of β-glucan and AX, with major differences in the extraction yield, while keeping a specific MWD. Of the tested parameters, the internal reactor pressure did not have any influence on the extraction yield and MWD of β-glucan and AX. Therefore, it is possible to use the autonomously regulated pressure, which does not exceed 20 bar; not only is this more cost effective, but it also requires lower safety standards. The interaction of temperature and time had the biggest influence in the quality of the produced BSG hydrolysates. Overall, the chemical structures (covalent and hydrogen bonds) of β-glucan and AX showed a thermo-labile behavior, when applying treatment temperatures between 130 and 200 °C. While little impact was observed below 140 °C, the rise in temperature up to 180 °C showed a significant decrease in MW with most of the yielded β-glucan and AX recovered in the low MW range (<10 kDa). Since temperatures above 160 °C additionally caused advanced saccharification and the formation of sugar degradation products, lower temperatures attracted greater interest. Finally, the treatment at 150 °C (5–15 min) resulted in the most efficient to reach high yields of β-glucan (>50 kDa) and AX (>20 kDa) that meet the literature-based conditions for potential human health effects without delivering too many undesired degradation products. HT will open numerous opportunities to reduce waste and find new applications for BSG in the food industry as potential sources for new food product development, like beverages or cereal-based products, as well as nutraceuticals.

## Figures and Tables

**Figure 1 foods-12-03778-f001:**
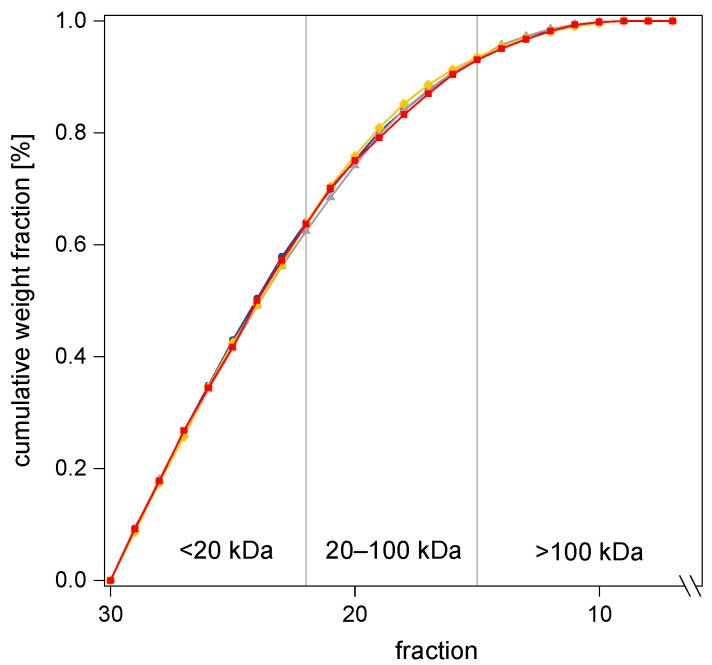
Cumulative weight fraction of MW [kDa] of AX in BSG hydrolysates obtained by HT, considering three grouped MW ranges (*n* = 3). Ramping times were excluded. Constant treatment temperature (160 °C), constant reaction time (10 min), and four different internal reactor pressures: ● < 20 bar, ▲ 50 bar, ◆ 100 bar, ■ 150 bar.

**Figure 2 foods-12-03778-f002:**
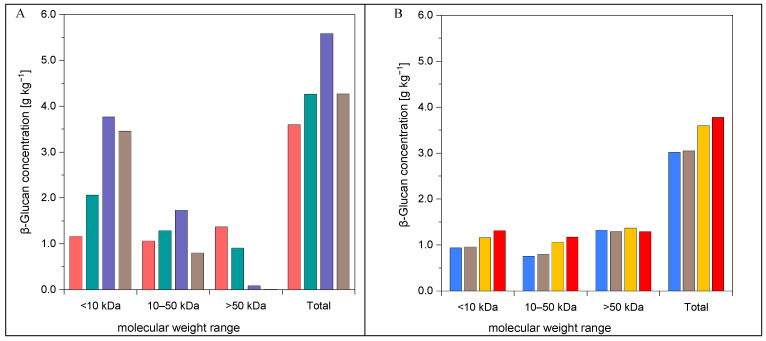
The β-glucan concentration [g kg^−1^] in BSG hydrolysates obtained by HT, determined within defined MW ranges (*n* = 3). Ramping times were excluded. Internal reactor pressure was <20 bar. (**A**) Constant reaction time (10 min) and four different treatment temperatures: ■ 140 °C, ■ 150 °C, ■ 160 °C, ■ 170 °C; (**B**) constant treatment temperature (140 °C) and four different reaction times: ■ 2 min, ■ 5 min, ■ 10 min, ■ 15 min.

**Figure 3 foods-12-03778-f003:**
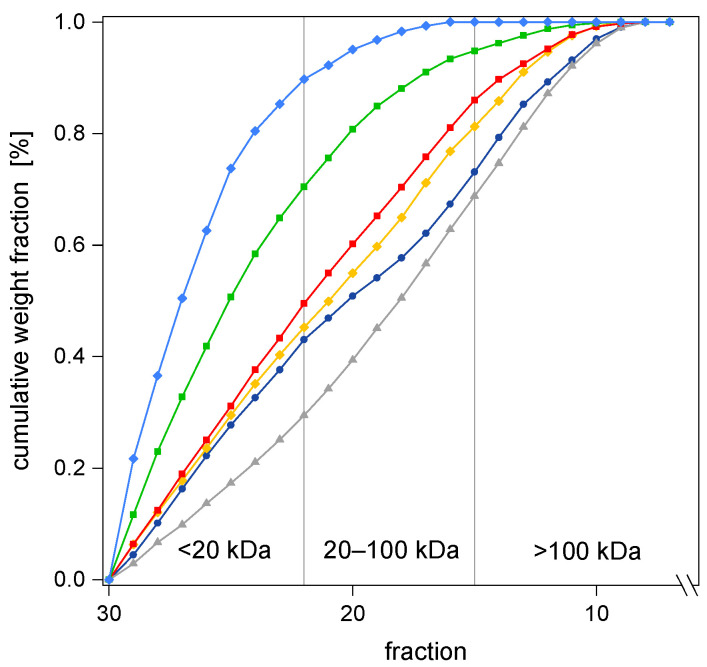
Cumulative weight fraction of MW [kDa] of AX in BSG hydrolysates obtained by HT, considering three grouped MW ranges (*n* = 3). Ramping times were excluded. Internal reactor pressure was <20 bar. Constant reaction time (2 min) and six different treatment temperatures: ● 130 °C, ▲ 140 °C, ◆ 150 °C, ■ 160 °C, ■ 170 °C, ◆ 180 °C.

**Table 1 foods-12-03778-t001:** Impact of internal reactor pressure at different temperature/time combinations on total concentration [g kg^−1^] of water-extractable β-glucan and AX in BSG hydrolysates (*n*=3).

Variable Parameters			Yield [g kg^−1^]				
Time	Pressure		β-Glucan			AX	
			160 °C	180 °C		160 °C	180 °C
**2 min**							
	<20 bar		6.3 ^z^	4.6 ^c,d^		73.0 ^v,s^	143.2 ^x^
	50 bar		6.1 ^z,g^	4.5 ^c,d^		63.3 ^v^	149.6 ^x,u^
	100 bar		5.7 ^g^	4.7 ^c,d^		70.5 ^v^	140.7 ^x^
	150 bar		5.9 ^z,g^	4.7 ^c,d^		78.9 ^s^	156.8 ^u^
**5 min**							
	<20 bar		5.7 ^g^	4.1 ^b,d,e^		84.1 ^s^	146.0 ^x^
	50 bar		5.4 ^c,g^	3.9 ^b,e^		69.8 ^v^	161.8 ^u^
	100 bar		5.6 ^g^	4.2 ^b,d,e^		72.7 ^v,s^	153.9 ^x,u^
	150 bar		5.8 ^g^	3.9 ^b,e^		76.3 ^v,s^	144.4 ^x^
**10 min**							
	<20 bar		5.6 ^g^	3.4 ^a,b,h^		104.5 ^r^	151.3 ^x,u^
	50 bar		5.7 ^g^	3.0 ^a,h,f^		89.8 ^r,s^	160.4 ^u^
	100 bar		5.3 ^c,g^	3.1 ^a,h,f^		90.1 ^r,s^	150.6 ^x,u^
	150 bar		5.3 ^c,g^	3.0 ^a,h,f^		91.5 ^r,s^	154.9 ^x,u^

Mean values in the same column followed by the same letter are not significantly different (*p* > 0.05).

**Table 2 foods-12-03778-t002:** Effect of treatment temperature and reaction time on total concentration [g kg^−1^] of water-extractable β-glucan and AX in BSG hydrolysates compared to the untreated BSG control sample (*n* = 3). Internal reactor pressure was <20 bar.

Variable Parameters			Yield [g kg^−1^]	
Time	Temperature		β-Glucan	AX
	no HT		0.2	2.1
**2 min**				
	130 °C		3.0 ^a,h,f^	16.8 ^y^
	140 °C		3.0 ^a,h,f^	16.8 ^y^
	150 °C		3.9 ^b,e^	37.4 ^w,t^
	160 °C		6.3 ^z^	73.0 ^v,s^
	170 °C		5.2 ^c,g^	150.8 ^x,u^
	180 °C		4.6 ^c,d^	145.6 ^x^
	200 °C		0.0	103.3 ^r^
**5 min**				
	130 °C		3.1 ^a,h,f^	20.8 ^y,w^
	140 °C		3.0 ^a,h,f^	23.1 ^y,w^
	150 °C		3.6 ^a,b^	52.9 ^t^
	160 °C		5.7 ^g^	85.5 ^s^
	170 °C		4.8 ^c,d^	154.1 ^x,u^
	180 °C		4.1 ^b,d,e^	146.0 ^x^
	200 °C		0.0	89.3 ^s,r^
**10 min**				
	130 °C		3.2 ^a,h^	20.9 ^y,w^
	140 °C		3.6 ^a,b^	29.4 ^w^
	150 °C		4.3 ^d,e^	66.5 ^v^
	160 °C		5.6 ^g^	104.5 ^r^
	170 °C		4.3 ^d,e^	162.3 ^u^
	180 °C		3.4 ^a,b,h^	151.3 ^x,u^
	200 °C		0.0	47.9 ^t^
**15 min**				
	130 °C		2.8 ^h,f^	32.4 ^w^
	140 °C		3.8 ^b,e^	31.2 ^w^
	150 °C		4.6 ^c,d^	74.6 ^v,s^
	160 °C		n.a.	n.a.
	170 °C		3.5 ^a,b^	178.3
**20 min**				
	150 °C		5.4 ^c,g^	86.6 ^s^
	160 °C		4.2 ^b,d,e^	139.6 ^x^
	170 °C		3.1 ^a,h,f^	102.2 ^r^
	180 °C		2.5 ^f^	89.4 ^r,s^
**30 min**				
	140 °C		3.4 ^a,b,h^	49.2 ^t^
	150 °C		3.7 ^a,b,e^	128.8
	160 °C		n.a.	n.a.
	170 °C		2.8 ^h,f^	100.8 ^r,s^

Mean values in the same column followed by the same letter are not significantly different (*p* > 0.05).

**Table 3 foods-12-03778-t003:** Proportional distribution [%] of water-extractable β-glucan in a BSG suspension on grouped MW ranges after HT at different treatment temperatures and reaction times (*n*=3). Internal reactor pressure was <20 bar.

Variable Parameters			β-Glucan Distribution [%] on MW Ranges			
Temperature	Time		<10 kDa	10–50 kDa	50–100 kDa	>100 kDa
140 °C	**10 min**		32.3	29.6	26.5	11.6
150 °C	**10 min**		48.4	30.3	18.1	3.2
160 °C	**10 min**		67.4	31.1	1.5	0.0
170 °C	**10 min**		80.9	18.8	0.3	0.0
**140 °C**						
	2 min		31.1	25.1	24.3	19.5
	5 min		31.4	26.2	27.0	15.4
	15 min		34.8	31.0	24.5	9.7
**150 °C**						
	15 min		53.1	30.0	15.3	1.6
	20 min		65.1	29.1	5.8	0.0
	30 min		70.9	25.7	3.4	0.0
**160 °C**						
	2 min		54.2	28.4	14.5	2.9
	5 min		65.5	25.9	8.4	0.3
	20 min		87.0	13.0	0.0	0.0

**Table 4 foods-12-03778-t004:** Proportional distribution [%] of water-extractable AX in a BSG suspension on grouped MW ranges and the respective yields [g kg^−1^] after HT at different treatment temperatures and reaction times (*n* = 3). Internal reactor pressure was <20 bar.

Variable Parameters			AX Distribution [%] on MW Ranges				Yield [g kg^−1^]	
Temperature	Time		<20 kDa	20–100 kDa	>100 kDa		<20 kDa	>20 kDa
140 °C	**10 min**		30.9	48.9	20.1		9.1	20.3
150 °C	**10 min**		47.7	36.0	16.3		31.7	34.8
160 °C	**10 min**		63.0	30.5	6.5		65.8	38.6
170 °C	**10 min**		78.5	19.4	2.0		127.4	34.9
**140 °C**								
	2 min		28.4	39.2	32.4		4.8	12.0
	5 min		30.8	43.9	25.4		7.1	16.0
	15 min		34.8	49.2	16.0		10.9	20.4
**150 °C**								
	2 min		44.3	35.9	19.8		16.6	20.8
	5 min		46.5	35.0	18.5		24.6	28.3
	15 min		49.0	35.5	15.5		36.5	38.1
	20 min		49.6	36.0	14.4		43.0	43.6
**160 °C**								
	2 min		48.5	36.3	15.2		35.4	37.6
	5 min		49.8	37.5	12.6		42.6	42.9
	20 min		66.4	30.3	3.2		92.7	46.8
**170 °C**								
	2 min		70.5	23.5	6.0		109.2	45.7
	5 min		74.9	21.4	3.7		115.3	38.7
	15 min		85.7	14.1	0.2		152.8	25.5
	20 min		89.1	10.9	0.0		91.1	11.1

**Table 5 foods-12-03778-t005:** Total concentration [g kg^−1^] of water-extractable β-glucan and AX in a BSG hydrolysates and proportional distribution [%] on grouped MW ranges and the respective yields [g kg^−1^] after HT at different treatment temperatures and reaction times (*n* = 3). Internal reactor pressure was <20 bar.

Variable Parameters		Total Concentration β-Glucan [g kg^−1^]	Distribution [%] on MW Ranges [kDa]					Yield [g kg^−1^]			
Temperature	Time		<10 kDa	10–50 kDa	50–100 kDa	>100 kDa		<10 kDa	10–50 kDa		>50 kDa
150 °C	20 min	5.4 ^g^	65.1	29.1	5.8	0.0		3.5 ^i^	1.6 ^k^		0.3 ^m^
160 °C	5 min	5.7 ^g^	65.5	25.9	8.4	0.3		3.7 ^i^	1.5 ^k^		0.5 ^m^

140 °C	10 min	3.6 ^a^	32.3	29.6	26.5	11.6		1.1	1.1 ^n^		1.4
150 °C	30 min	3.7 ^a^	70.9	25.7	3.4	0.0		2.6	1.0 ^n^		0.1
**Variable Parameters**		**Total Concentration** **AX [g kg^−1^]**	**Distribution [%] on MW Ranges [kDa]**					**Yield [g kg^−1^]**			
**Temperature**	**Time**		**<20 kDa**	**20–100 kDa**		**>100 kDa**		**<20 kDa**		**>20 kDa**	
150 °C	20 min	85.5 ^s^	49.6	36.0		14.4		43.0 ^p^		43.6 ^p^	
160 °C	5 min	86.6 ^s^	49.8	37.5		12.6		42.6 ^p^		42.9 ^p^	

160 °C	10 min	104.5 ^r^	63.0	30.5		6.5		65.8		38.6	
170 °C	20 min	102.2 ^r^	89.1	10.9		0.0		91.1		11.1	

Mean values [g kg^−1^] in the same column followed by the same letter are not significantly different (*p* > 0.05).

**Table 6 foods-12-03778-t006:** Effect of treatment temperature and reaction time on HMF and furfural concentration of BSG hydrolysates in comparison to the untreated control sample (*n* = 3). Hydrolysates with variable temperature/time combinations had a constant internal reactor pressure <20 bar.

		HMF [mg L^−1^]							Furfural [mg L^−1^]					
		Tolerable Daily Intake: 132 mg [48]							Acceptable Daily Intake: 0.5 mg/kg bw [50]					
Variable Parameters		Time in min							Time in min					
		2	5	10	15	20	30		2	5	10	15	20	30
Temperature		Control sample: 0.2 mg L^−1^							Control sample: 0.0 mg L^−1^					
130 °C		0.6 ^a^	0.5 ^a^	1.2 ^a^	1.3 ^a^	n.a.	n.a.		0.1 ^k^	0.1 ^k^	0.2 ^k^	0.2 ^k^	n.a.	n.a.
140 °C		1.0 ^a^	1.0 ^a^	1.6 ^a^	1.9 ^a^	n.a.	n.a.		0.1 ^k^	0.1 ^k^	0.2 ^k^	0.1 ^k^	n.a.	n.a.
150 °C		1.9 ^a^	1.9 ^a^	2.4 ^a^	2.6 ^a^	6.8 ^b^	17.7 ^e,f^		0.3 ^k^	0.3 ^k^	0.5 ^k^	0.7 ^k^	1.3 ^l^	3.0 ^o,p^
160 °C		6.1 ^b^	6.3 ^b^	7.3 ^b,d^	n.a.	10.9 ^c^	n.a.		1.2 ^l^	1.3 ^l^	1.4 ^l,n^	n.a.	2.8 ^m^	n.a.
170 °C		8.4 ^b,c^	9.7 ^c,d^	10.7 ^c^	15.4 ^e^	24.7 ^g^	37.8 ^i^		1.6 ^l,m^	2.0 ^m,n^	2.3 ^m^	4.2 ^o^	8.7 ^q^	11.7 ^s^
180 °C		15.0 ^e^	17.6 ^e,f^	20.5 ^f^	n.a.	39.0 ^i^	n.a.		2.7 ^o^	2.7 ^o,p^	3.7 ^p^	n.a.	10.3 ^s^	n.a.
200 °C		23.9 ^g^	26.8 ^g,h^	28.1 ^h^	n.a.	n.a.	n.a.		5.8 ^q^	7.7 ^q,r^	9.7 ^r^	n.a.	n.a.	n.a.

Mean values in the same column followed by the same letter are not significantly different (*p* > 0.05).

## Data Availability

Data are contained within the article.

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
