# Peer review of "Influence of Hydrothermal Treatment of Brewer’s Spent Grain on the Concentration and Molecular Weight Distribution of 1,3-1,4-β-D-Glucan and Arabinoxylan"

_foods, 2023, doi:10.3390/foods12203778_

Round 1
Reviewer 1 Report
The evaluated work presents interesting results of hydrothermal treatment (HT) of brewer’s spent grain to recover of potentially pro-healthy 1,3-1,4-β-D-glucan and arabinoxylan fractions. The HT process could be considered as a green chemistry solution for recovery of new food products or nutraceuticals from various agri-food wastes. Various parameters of the HT process were tested to know how they affect the distribution of thermal degradation products of the indicated polysaccharides. The authors particularly paid attention to the formation of glucan and arabinoxylan fractions with potential health-promoting effects. Before publication, the work requires some corrections, as detailed below.
DETAILED COMMENTS
Line 99: “Numerous investigations have shown …” – could you provide some examples, there is no citation in this section
Lines 107-125: the purpose of the study was described in too much detail. I suggest to transfer some information to Methodology section (point 2.2. Equipment for HT and test design), for example about the selection of specific thermal treatment parameters.
Line 144: sampling could be conducting ? -> sampling was… - in general the manuscript requires standardization of the past tense
Lines 154-156: “For a large part of treatments, the internal reactor pressure is not set, but adjusted itself autonomously at a function of temperature in a closed system. Additionally, experiments with manually set pressure levels (50, 100, 150 bar) are also carried out.” - it is not clear how the conducted research was handled -> see: ….is not set, are also carried out… - the past tense should be used
Line 162:
- “pure β-glucan model solutions” – model or standard solutions?;
- “suspensions usually comprising 3% (w/w) solids” – what does it mean/ why - > usually?
Lines 164-165: “samples may be gently centrifuged to achieve a faster sedimentation” – what procedure was applied in case of the study? -> may be or were?
Lines 169-179: please provide information about the origin of the enzymes and their activity, concentration, type of buffer solution of pH 6.5 that were used. Additionally, I propose to standardize the spelling to the past tense (within whole text), eg. samples are suspended à were suspended
Line 202: “model substances” – standards?
Lines 207-208: what concentration of samples was applied?
Line 223: were the samples injected directly or after dilution?
Lines 215-227: in what concentration range HMF and furfural were analyzed ?
Line 234: model solution or standard one?
Lines 235-236: “Quantitative Initial assessment on the MWD was done using single substance model 235 solutions; fragmentation and decomposition patterns of the individual substances could be determined.” – please, correct the style
Line 242-243: dropping to zero when exceeding 200 °C? – was applied temperature higher than 200 °C?
Line 252: the process of auto-hydrolysis should be explained, what is it mechanisms in case of studied samples
Line 255: that has not yet been subjected to HT – why yet? It was a control sample
Line 266: the sentence requires clarification -> what samples it is about
Line 274: please correct the way of citation (number of authors)
Line 298: model substances -> standards?
Line 301: concentration of which analytes?
Lines 311-312: Benito-Román, Alonso, 311 Gairola and Cocero [30] – please, correct the way of citation
Line 332: “a clear reduction to 4.2 g kg-1…” – please add-> in β-glucan yield
Lines 341, 352, 408, 436, 455: Xu, Xu and Zhang [32] – please correct citation
Figure 1: the colors in the drawing and in the legend are different
Lines 500-501: “Regarding the area in between (20–100 kDa), the time dependent correlation changes in accordance with the treatment temperature.” – correct the sentence -> no verb
Line 530: “3.3.3. Importance of MWD on beneficial effects of β-glucan and AX on human health” – this aspect was no experimentally studied. Additionally, Tabel 6 presents the same results as were included in table 3, however selected. Since the authors did not carry out research that would justify the title of section 3.3.3 as quoted above, and Table 6 contains previously presented data, the discussion of the results from this part should be included in the previous sections or constitute some kind of summary for selected parameters that allowed to achieved fractions potentially important for human health.
Line 551: “identical total concentration” – concentration of what?
Lines 567-568: Carvalheiro, Duarte, Lopes, Parajó, Pereira and Gı́rio [49] – way of citation
Line 570: “that support potential human” - The authors do not present the results of in vivo/in vitro tests that would confirm the activity of the fractions they obtained. I therefore consider such statements to be unconfirmed. We can only say about potential health effect.
Lines 620-621: (see Table 620 6Fehler! Verweisquelle konnte nicht gefunden werden.) - ?
General note to the discussion section: the discussion is very detailed, making it difficult to follow. The authors should try to make it more synthetic.
It is recommended to have the text linguistically proofread before publication.
Author Response
Please see the attachment.
Best
Julia Steiner

Reviewer 2 Report
Materials:
1) Please provide, the temperature choice of 70 C for the fresh BSG drying.
2) please provide experiments design for better understanding of research steps.
Results:
1) Discussion part is practically absent in chapter 3.2. plaase add a little.
2) The importance of MWD on beneficial effects of β-glucan and AX on human health will be provided only by clinical research. Was any clinical studies provided in the present research?
3) Some misunderstandings take place at rowas 620 and 621.
Conclusions:
You can't talk about human effect without clinical studies, it is not correct.
Author Response

(The authors gave the same response as above.)
